# Advances and Trends in Pediatric Minimally Invasive Surgery

**DOI:** 10.3390/jcm9123999

**Published:** 2020-12-10

**Authors:** Andreas Meinzer, Ibrahim Alkatout, Thomas Franz Krebs, Jonas Baastrup, Katja Reischig, Roberts Meiksans, Robert Bergholz

**Affiliations:** 1Department of General Visceral, Thoracic, Transplant and Pediatric Surgery, UKSH University Hospital of Schleswig-Holstein Kiel Campus, Arnold-Heller-Strasse 3, 24105 Kiel, Germany; andreas.meinzer@uksh.de (A.M.); Jonas.baastrup@uksh.de (J.B.); katja.reischig@uksh.de (K.R.); roberts.meiksans@uksh.de (R.M.); 2Department of Obstetrics and Gynecology, UKSH University Hospital of Schleswig-Holstein Kiel Campus, Arnold-Heller-Strasse 3, 24105 Kiel, Germany; Ibrahim.alkatout@uksh.de; 3Department of Pediatric Surgery, Ostschweizer Children’s Hospital, Claudiusstrasse 6, 9006 St. Gallen, Switzerland; thomas.krebs@kispisg.ch

**Keywords:** pediatric surgery, minimally invasive surgery, fetal surgery, single-incision surgery, surgical techniques, surgical devices, open surgery, endoscopy, endoscopic surgery

## Abstract

As many meta-analyses comparing pediatric minimally invasive to open surgery can be found in the literature, the aim of this review is to summarize the current state of minimally invasive pediatric surgery and specifically focus on the trends and developments which we expect in the upcoming years. Print and electronic databases were systematically searched for specific keywords, and cross-link searches with references found in the literature were added. Full-text articles were obtained, and eligibility criteria were applied independently. Pediatric minimally invasive surgery is a wide field, ranging from minimally invasive fetal surgery over microlaparoscopy in newborns to robotic surgery in adolescents. New techniques and devices, like natural orifice transluminal endoscopic surgery (NOTES), single-incision and endoscopic surgery, as well as the artificial uterus as a backup for surgery in preterm fetuses, all contribute to the development of less invasive procedures for children. In spite of all promising technical developments which will definitely change the way pediatric surgeons will perform minimally invasive procedures in the upcoming years, one must bear in mind that only hard data of prospective randomized controlled and double-blind trials can validate whether these techniques and devices really improve the surgical outcome of our patients.

## 1. Introduction

Minimally invasive pediatric surgery has developed rapidly in the last 30 years extending from fetuses to 17 year old overweight adolescents. As many meta analyses and analyses of meta analyses comparing minimally invasive surgery to open procedures can be found in the recent literature, this review focuses on the current trends and advances which already have or may have an impact on pediatric minimally invasive surgery [1,2].

## 2. Materials and Methods

The techniques and development of minimally invasive pediatric surgery are reported descriptively. Print and electronic databases including Index Medicus, MEDLINE^®^/PubMed, EMBASE^®^, Cochrane Register (CENTRAL) and www.clinicaltrials.gov using the keywords or medical subject headings (MeSH) “laparoscopy,” “thoracoscopy”, “minimally invasive surgery”, “single incision laparoscopic surgery” “robotic surgery” and “pediatric surgery” were systematically searched. Latest entry was set to November 2020 and the search was not restricted to specific languages. We then conducted a cross-link search with references found in the literature. Full-text articles were obtained for potentially eligible publications and quality and eligibility criteria applied independently. Potential disagreements regarding the quality of studies and inclusions/exclusions were resolved by discussion. Articles were translated if needed and appropriate.

## 3. Review

### 3.1. Milestones of Pediatric Minimally Invasive Surgery

Minimally invasive surgery (MIS) evolved rapidly during the last 30 years, starting in adult general and gynecologic surgery. The first MIS appendectomy was performed by Kurt Semm, a gynecologist, which gained much attraction and dispute by the general surgical community [3]. The first laparoscopic cholecystectomies were performed in 1985 in Germany by Erich Mühe and in 1987 in France in a shared gynecology and general surgery practice by Phillipe Mouret [4,5]. Thereafter, especially laparoscopic cholecystectomy became the new minimally invasive standard of surgical care, although hard long term data were not yet available.

With the advent of electronic videoscopes, small instruments and insufflators feasible for children, MIS was also gaining ground in pediatric surgery (Figure 1).

Earlier, explorative laparoscopy (then called peritoneoscopy and performed by direct vision through the laparoscope) was performed in children as early as in 1971 by Gans for exploration of the abdomen through and small series of explorative peritoneoscopy were reported in by Carnevale for abdominal trauma in 1977 [6,7]. In 1976 Rodgers reported thoracoscopy for diagnostic reasons in children [8].

Concerning pediatric procedures, consecutively the first repair of a congenital diaphragmatic hernia (CDH) [9], the trans-thoracic approach to CDH [10], the first MIS repair of esophageal atresia [11], the Nuss repair of pectus excavatum [12], laparoscopic choledochal cyst excision [13] and the minimally invasive Kasai procedure were reported [14]. With the availability of robotic surgery in the early 2000s, some centers established robotic pediatric surgery programs [15]. As MIS advanced into the new millennium, new techniques as single incision laparoscopic surgery (SILS) were developed: Muensterer reported the first single incision procedures in infants in 2010 and his colleague Hansen a large series of 224 cases in 2011 [16,17].

### 3.2. Technical Developments of Pediatric Laparoscopy

The successful progress minimally invasive pediatric endoscopic surgery has experienced in the last 20 to 30 years has been fundamentally due to the incessant achievement of highly sophisticated technological equipment and thus the continuous development of instruments designed specifically for these surgical techniques. Equipment and instruments are designed to allow safe access to the child’s small anatomic cavity, maintain a good working space and perform maneuvers with the same or even better safety and efficacy as in open procedures.

While 10 mm-diameter rigid videoscopes and 5 mm-diameter instruments have been of use very commonly for decades, technical advances led to a widespread introduction of even smaller equipment more feasible in pediatric patients. Today there is a large choice of 3 mm diameter instruments by several companies such as Karl Storz (Tuttlingen, Germany), Wolff (Knittlingen, Germany) or Aesculap (Braun, Tuttlingen, Germany). Handling of tissues is very delicate and scarring is very satisfactory. More recently, microlaparoscopy with 2 mm diameter instruments has been introduced into pediatric surgery [18]. Although their use has its limitations due to fragility, the tendency for bending and difficulty in grasping there are selected and distinctive indications these instruments are applied for. Successful reports of thoracoscopic congenital diaphragmatic hernia repair in newborns, hiatoplasty with repair of an upside-down-stomach, laparoscopically assisted pull-through for Hirschsprung’s disease and laparoscopic transperitoneal pyeloplasty also suggest the further consideration of microlaparoscopy for advanced procedures in children [19].

The latter has also been accomplished by the introduction of smaller and distinguished devices for safe and meticulous hemostasis. More recently developed 5-mm-clips and -staplers and also advanced energy source devices, such as the LigaSure™ or the EnSeal^®^ have proved to be imminently helpful in minimally invasive pediatric surgery [20,21,22].

With the use of new-generation videoscopes and cameras that allow three-dimensional procedures together with a high-definition video format, such as the 4 mm Karl Storz IMAGE1 S™ 3D, vision in small and challenging spaces has improved immensely and therefore augmented the safety of the procedures (Figure 2) [23].

### 3.3. Natural Orifice Transluminal Endoscopic Surgery (NOTES)

NOTES promises to avoid the access through the abdominal cavity by introducing flexible or rigid instruments and optics through natural orifices as the stomach, vagina or rectum. By perforating these organs the instruments can be passed into the abdominal cavity for various surgical procedures. NOTES appears to offer no visible scars and promises lesser pain and thus faster recovery—including the inherent risk of elective viscerotomy and closure at the end of the procedure.

The first pure NOTES transvaginal cholecystectomy was performed by Tsin at Mount Sinai Hospital of New York in 2003 [24]. Since then, several pure NOTES procedures have been reported, as transvaginal nephrectomy or transvaginal appendectomy [25,26,27].

Pure NOTES is technically challenging, due to instrument clashing, suboptimal exposure and inline placement of the instruments compared to triangulation in laparoscopic surgery [28]. Hybrid NOTES, with an additional port through the umbilicus, appeared to increase instrumentation and safety, as the entry through the vagina or stomach can be visually controlled. Several reports found hybrid NOTES to be applicable in general surgery, as for cholecystectomy and appendectomy and several technical reports on instrumentation for NOTES can be found (Figure 3 and Figure 4) [29,30].

Bulian reported a prospective, randomized, nonblinded, single-center trial comparing hybrid NOTES to laparoscopy with 3mm instruments: Comparable in terms of safety, NOTES appeared to cause less pain, increase satisfaction with the esthetic result and improved postoperative quality of life in the short term [32,33]. A current meta-analysis concluded, that transvaginal NOTES cholecystectomy, adnexectomy, and appendectomy appears safe and truly minimally invasive [34].

NOTES has, to the best of our knowledge, been reported only once in the pediatric population: Lamas-Pinheiro reported a hybrid natural orifice transluminal endoscopic surgery approach for laparoscopic Duhamel procedure in three cases. The 12 mm port for the endoscopic stapler was introduced through the rectum instead of inserting it transabdominally (Figure 5) [35].

Nevertheless, transgastric endoscopic drainage of pancreatic pseudocysts after blunt abdominal trauma and pancreatitis is common in children and counts, technically, into the armamentarium of NOTES. Drainage can be achieved by placing a stent or leaving the gastric incision open to the pseudocyst for drainage and secondary healing [36,37]. Therefore, one can postulate that NOTES is already in clinical application in pediatric minimally invasive surgery.

### 3.4. Single Incision Laparoscopic and Thoracoscopic Surgery

As previously stated technology and innovation continue to advance the field of pediatric MIS thus creating space for the establishment of single-incision laparoscopic surgery (SILS) and uniportal video assisted thoracoscopic surgery (VATS). Applications have been previously described using this approach for various general, urologic, thoracic and pediatric surgical procedures. For example, there are reports of successful surgical treatment of acute appendicitis, gastroesophageal reflux, ureteropelvic junction stenosis, and pleural empyema [38,39,40,41].

Despite a lack of literature promoting this surgical approach across the pediatric population there are a few randomized studies that support single-incision techniques for appendectomy in both feasibility and safety compared to conventional laparoscopic surgery [42,43].

As some disadvantages have been generally attributed to laparoscopic procedures, such as higher costs [44], longer operative time [45], and more demanding surgical skills and equipment [46], transumbilical laparoscopic-assisted appendectomy (TULAA) has emerged as a variation from the standard laparoscopic appendectomy [47]. It combines the advantages of laparoscopy and open surgery, with global visualization of the abdominal cavity and minimal invasion with lower costs and instrumental requirements [48,49,50,51].

Based on those available data, single incision laparoscopic or laparoscopic assisted surgery has for many pediatric surgeons emerged as the first choice for the mode of minimally invasive access in many different procedures. Conversion to classical three port laparoscopic or thoracic surgery can be easily performed when procedures were started through a single incision. Therefore, the single incision procedure is a versatile technique in children, providing a safe, effective, and the least invasive treatment for different diseases [52].

### 3.5. Endoscopic Pediatric Surgery

With the development of endoscopy, more pathologies become addressable by newly designed devices. For achalasia, Heller’s myotomy has been the standard of care [53]. Per-oral endoscopic myotomy (POEM), a modern treatment for achalasia, has only recently emerged as an option for pediatric patients. Wood reported in her study on 21 pediatric cases POEM to be a viable and safe treatment for pediatric patients with achalasia [54].

Appendicitis is the most common abdominal emergency for surgery in childhood. Appendectomy can be performed by the open approach, classical three port laparoscopic or single port procedures. A new technique is endoscopic retrograde appendicitis therapy (ERAT). It consists of five steps, which are all performed endoscopically after insertion of the colonoscope into the cecum and identification of the appendiceal orifice: endoscopic appendiceal intubation, appendiceal decompression, retrograde appendicography, stent drainage and cleansing of the appendiceal lumen. First described by Liu in 2016, Kang et al. reported in November 2020 their series of 36 children treated with ERAT in a randomized prospective trial. They concluded that ERAT appears to provide a new alternative to surgical appendectomy for uncomplicated appendicitis in children [55,56].

### 3.6. Robotic Pediatric Surgery

In pediatric surgery, many procedures are restricted by the limited working space of the small abdominal and thoracic cavity, encumbering even 3-mm instrument and multi-port procedures. A further development of the minimally invasive technique is robot or computer-assisted surgery, in which the instruments inserted into the body are remotely controlled by the surgeon, who is placed at a console next to the patient or even far more remote.

Due to the magnification of the operative field, application of 3D technology and thus spatial vision, improved ergonomics for the surgeon and a greater range of motion of the robotic instruments compared to conventional laparoscopic instruments, robotic assisted minimally invasive surgery appears to be beneficial over conventional minimal invasive surgery, especially in complex reconstructive procedures [57,58,59,60,61].

Currently, there are two systems commercially available and certified for robotic surgery in children: the DaVinci (Intuitive Surgical, since 2001) and Senhance (Transenterix, since 2020) robotic system.

The DaVinci robotic system includes a control unit for the surgeon and a patient side cart with four remotely controlled arms. To each arm, a camera with 3D Vision, different surgical instruments and energy or stapling devices for vessel sealing and dissection can be attached. The diameter of the instruments is 8 mm and their tip is bendable with seven degrees of freedom analogous to the human wrist (“endowrist”). Smaller diameter instruments (5 mm) are available too, but due to their angulation of the tentacle-like continuum tool shafts rather than the articulated wrist joints that characterize standard 8-mm instruments, the smaller 5-mm instruments have less dexterity than the standard 8-mm instruments in spatially constrained operative fields (Figure 6) [62].

Thakre examined the feasibility of the DaVinci robotic surgery in small cavities and reported that surgical drills could only be performed in cubes with edges of 70 mm length or greater [63]. This impairment in small cavities is a major limitation of the DaVinci surgical system in small cavities, such as in newborns and infants [64,65]. Although sporadic reports exist on robotic infant surgery, the DaVinci is mainly used in older children [15].

The second robotic system commercially available and certified for application in children larger than 10 kg of body weight is the Senhance (Transenterix). This system consists of a control unit for the surgeon and three to four separate carts, each with one arm for either camera or instruments. The instruments resemble classic 5 mm diameter laparoscopic instruments. In contrast to the DaVinci system, the instruments are not articulating, except an 8 mm diameter articulating needle driver, but offer haptic force—feedback. Additionally, a complete range of 3 mm diameter instruments is also available. As smaller diameter instruments can be placed more closely together and do not need a long insertional depth, it may be hypothesized that robotic surgery might be feasible in small cavities with this system, in contrast to the DaVinci (Figure 7).

Due to its relatively new emergence on the market, not much data can be found on potential feasibility of the Senhance in small cavities: It was demonstrated in inanimate models, that even in small volumes of 90 mL (edges of 2.9 cm × 6.3 cm × 4.9 cm boxes) intracorporal suturing and manipulation appears feasible with this system [66]. Currently, the first pediatric robotic procedures have been performed in the Department of Pediatric Surgery at the Maastricht University Medical Center+ [67].

Also being counted as robots are automated suturing robots, like the KidsArm, an image-guided pediatric surgical robot, to automate anastomosis, which has been reported in 2013 or the STAR reported by Leonard in 2014, both awaiting wider examination by pediatric surgeons [68,69].

### 3.7. Fetal Surgery

Fetal surgery is pediatric surgery and pediatric surgery is fetal surgery: Since congenital conditions and malformations are often leading to serious consequences on fetal and eventually children’s development the field of fetal surgery has grown to be of major interest for pediatric surgeons from the early 1980s, with Michael Harrison being the most prominent innovator in this field [70,71].

Today, prenatal diagnostics allow for a high rate of fetal anomaly detection from a very early gestational age. This allows for an early-stage multidisciplinary approach for fetal therapy, joining the expertise of various specialists. The surgeon necessarily must rely on neonatologists, anesthesiologists, radiologists and obstetrician—gynecologists among many others, to contribute to the successful treatment of the fetus.

Improvement in pathophysiological knowledge and the development of therapeutic tools led to advancement in fetal surgery and set in motion changes in treatment approaches from open procedures to fetoscopic techniques for many conditions of the unborn child [72].

With selective fetoscopic laser photocoagulation a Diode or Nd:YAG Laser is used to treat twin twin transfusion syndrome (TTTS) successfully and evidentially ameliorates the double-twin survival rate [73].

Fetoscopic endoluminal tracheal occlusion (FETO) by fetal endotracheal balloon placement for isolated severe congenital diaphragmatic hernia (CDH) improves neonatal survival significantly [74], while being subject to ongoing investigation in an international randomized trial called The Tracheal Occlusion To Accelerate Lung (TOTAL) growth trial (www.totaltrial.eu) for severe and moderate pulmonary hypoplasia [75].

Fetal cystoscopy is used to treat lower urinary tract obstructions in most cases due to posterior urethral valves, for which Ruano et al. found a significant improvement in survival at 6 months after intervention and an advantage of fetal cystoscopy for renal function [76].

While fetoscopic myelomeningocele (MMC) repair-techniques showed disadvantages especially in safe closure of the MMC defect comparable to open repair [77], Patel et al. just recently presented a promising fetoscopic multilayer closure with dural patch repair using a standardized, 3-port, carbon dioxide insufflation technique for the intrauterine treatment of MMC [78]. Furthermore, fetoscopic bimanual surgery is associated with a higher risk of premature rupture of membranes—caused by chorioamniotic separation when compared to open surgery as the uterotomy is stapled and the membranes are fixed to the uterine wall in contrast to the fetoscopic ports which are inserted by puncture. Michael A. Belfort developed a hybrid open/fetoscopic method to lower the risk of preterm rupture of amniotic membranes anchoring them to the uterine wall without opening the uterus by hysterotomy and thus pushed fetoscopy and fetoscopic bimanual surgery to another limit [79,80,81].

Altogether, fetoscopy is effective for treating several fetal anomalies at present. In the future continuous refinement of the techniques and technologic advances will allow the use of fetoscopy more extensively and aid entry to treatment for other pathologies, such as in utero gastroschisis repair, for carefully selected fetuses [72,82,83,84].

### 3.8. Outlook: The Future of Pediatric Minimally Invasive Surgery

Technical development is unstoppable, new and smaller instruments, devices and systems are emerging on an ever growing market of miniaturizing and digitizing surgery [85]. Some devices only exist in the heads and minds of pediatric surgeons, others have already found their way into preclinical or even clinical evaluation:

#### 3.8.1. Magnetic Anastomosing Devices

In surgery, one of the most critical parts is forming a new connection between hollow organs, vasculature or nerve fibers, called anastomoses. Classically, those connections were sutured by hand which takes a relevant amount of operating room time and every surgeon knows the dreaded feeling when suspecting his or her anastomosis to become insufficient, which implies possible severe consequences for the patient. With the development of anastomosing devices for intestinal anastomoses, called intestinal staplers, the time to perform an anastomosis could drastically be reduced and also standardized, as every type of stapler works the same [86]. Although some staplers can be applied endoluminally, they still need a laparotomy or laparoscopy for visual control and firing. A pure endoscopic application might be the use of magnets—a device that could automatically and consistently produce an optimal anastomosis, reduce morbidity and save considerable operative time and resources. Two magnets are specifically used to perform an anastomosis by compressing the according intestinal wall between each other: the tissue between the becomes ischemic and sloughs while the outer rim heals, thus establishing the anastomosis. Once the anastomosis is complete, the two magnets would be automatically transported through the intestine by peristalsis [87,88].

Different types of magnets and techniques have been examined in experimental and human settings, especially for esophageal atresia or esophageal strictures [89,90,91,92].

Research culminated in the development of an FDA cleared device for endoscopic magnetic anastomosis in infants with esophageal atresia (Figure 8) [93,94,95].

#### 3.8.2. Articulating Laparoscopic Instruments and Devices for Single-Incision Laparoscopic Surgery

One disadvantage of classical laparoscopic surgery is the straight instruments. Therefore, the success of the procedure depends on whether the surgeon is able to place the instruments in a certain angle which allows him access to all areas of the operative field but also sufficient angulation of the instruments to each other, especially for suturing and knot tying in reconstructive procedures. One advantage of robotic surgery is the application of angled or wristed instruments, which give the surgeon up to seven degrees of bendable or rotational freedom, just like having his or her hands with the articulating wrist inside of the patient. By many surgeons, these wristed or articulating instruments are deemed as one major benefit of robotic surgery.

In the last years, articulating instruments also became available for laparoscopic surgery. The FlexDex device is a laparoscopic needle driver with the same degrees of wristed angulation as offered in the DaVinci robotic system [97,98,99]. It has been clinically evaluated in children and appears to improve reconstructive procedures without the costs as for a robot (Figure 9) [100].

Several other articulating laparoscopic instruments, or prototypes of, have been either FDA approved, evaluated in dry lab trainer or live animal models with ambiguous results towards feasibility and learning curve: The Radius Surgical System (Tübingen Scientific) was evaluated in experimental and clinical settings and appeared to improve intracorporal maneuverability [101,102]. The Artisential Laparoscopic System (Livsmed) is FDA approved and offers a wide range of articulating instruments as well as energy devices [103,104]. The Hand-X electronic articulating needle driver (Human Extensions, Netanya, Israel) received FDA approval in 2018 and offers a 5 mm diameter wristed electronically driven instrument [105]. The SymphonX Surgical Platform (Fortimedix Surgical B.V.) received FDA approval on 26 August 2016. According to the Society of American Gastrointestinal and Endoscopic Surgeons (SAGES), “it provides a path of entry for laparoscopic instruments and camera through a single site and allows for triangulation similar to standard laparoscopy. The device fits through a standard 15-mm trocar and has 4 channels, enabling a surgeon to use two 5-mm instruments, a 5-mm camera and a 3-mm device. The device does not require inversion or hand crossing to achieve triangulation” [106]. It was also recently evaluated during human application in adult general surgery [107,108].

The Spider Surgical system (Single Port Instrument Delivery Extended Reach, Transenterix, Durham, NC, USA) is a platform similar to the SymphonX for single port procedures. It has been evaluated in different human general surgical, gynecologic and urological procedures and appeared feasible and safe [109,110,111,112].

Several other articulating instruments as well as prototypes have been described and examined during the last ten years of development, which was mainly driven by the push of Intuitive’s DaVinci endowrist and robotic surgery [103,104,113,114,115,116,117,118].

Another approach is followed by Microsure (Eindhoven, The Netherlands), which developed the MUSA, a robot for open microsurgical procedures such as vessel or nerve anastomosis. This system has recently been evaluated in human gynecological surgery and appears to confer the feasibility of connecting vessels with a diameter between 0.3 and 0.8 mm for the reconstruction of lymphatic flow and vascularized tissue transplantation [119,120,121].

All those abovementioned devices promise an improvement in the surgical care of pediatric patients, nevertheless, none of those has yet been systematically evaluated for its probable application in pediatric surgery.

#### 3.8.3. New Robots for Children

Until this review, two robotic systems are commercially available for application in children. Both exhibit specific advantages and disadvantages as described above. With more emerging robotic systems appearing on the market in the upcoming years, the anticipation of pediatric surgeons increases, for a system that offers full intracorporal maneuverability with fully wristed instruments that are less than 5 mm in diameter and therefore applicable in children as small as newborns and infants.

The Dexter (Distalmotion, Lausanne, Switzerland) offers fully wristed 8 mm robotically instruments remotely controlled from a console similar to the Da Vinci system. This system consists of the robotic instruments only, without a camera and optical console, therefore being cheaper than the Da Vinci and because of its reduced size fitting into the setting of classical laparoscopic surgery enabling the surgeon to instantly switch to the laparoscopic robotically assisted part of the procedure. With its 8 mm diameter instruments its application in pediatric surgery in small children has to be critically evaluated [122].

Verb Surgical (Santa Clara, CA, USA), a cooperation of Google and Johnson & Johnson appears to develop a robotic system that will be integrated in a more comprehensive pre- and postoperative setting with enhanced medical data science [123]. Whether this system will be applicable in children, has to be evaluated.

Avatera (avateramedical, Leipzig, Germany) is a robot conceptually similar to Intuitive’s system but offers 5 mm diameter fully wristed seven degrees of freedom instruments with less angulation than the 5 mm instruments of the Da Vinci. Whether this will be an advantage in small cavities, such as in small children and infants, will have to be critically evaluated [124].

CMR Surgical (Cambridge, UK) developed the Versius robotic system, which has already been clinically applied and evaluated in general surgical, urological and gynecological procedures [125,126,127,128,129,130]. With its 5 mm diameter fully wristed instruments its application in pediatric surgery appears promising. Any preclinical or clinical evaluation of its feasibility in children and small infants is pending.

In conclusion, there are many more robotic surgical systems either already in the market or emerging in the upcoming years. Whether they will be feasible, safe and therefore applicable in pediatric surgical procedures has to be critically evaluated. Pediatric surgeons should be encouraged to participate in this process in order to give their future patients probable access to this rapidly evolving technology.

#### 3.8.4. Deployable Minirobots

Another development of robotic surgery is the idea of deployable minirobots which can be inserted into the abdominal or thoracic cavity and perform surgical tasks by remote control. Therefore, multiple minirobots can be deployed by just one small incision, further reducing the operative trauma, and may provide task assistance without the constraints of the entry incision.

Although a concept more appearing as science fiction, some groundbreaking work has already been reported: Forgione was able to deploy a remotely controlled instrument with lighting, camera or graspers, assisting in cholecystectomies in an animal model [131]. Shah reported a multiarmed dexterous miniature in vivo robot with stereovision, graspers and cautery (University of Nebraska AB1 Robot) presented by Lehman in 2008 [132]. This robot was successfully applied for assisting various surgical procedures in animal models. In the future, deployable and remotely controllable surgical devices will allow us to perform procedures with fewer incisions that we cannot do today with conventional minimally invasive techniques. Therefore, the future of true minimally invasive surgery has not arisen yet.

#### 3.8.5. Hybrid Procedures

The concept of providing endoscopic assistance for open or laparoscopic surgical procedures is not new, but has not found its way into clinical application in pediatric surgery, although some pioneering and groundbreaking work has already been presented. Laparoscopic endoscopic cooperative surgery (LECS) has been evaluated and held to be a feasible technique for surgery in the upper gastrointestinal tract [133,134]. The case of a 17 year old pediatric patient with non-exposed endoscopic wall-inversion surgery for a gastrointestinal stromal tumor was reported by Matsumoto [135].

In pediatric surgery, the modern approach to an anorectal malformation without a fistula is the posterior sagittal anorectoplasty, as described by Alberto Pena [136]. Although accepted by the pediatric surgeons, the operation consists of surgically splitting the remains of the anal sphincter muscle complex to identify the rectum. A new approach was suggested by Muensterer, the endoscopic assisted posterior anorectoplasty (ePARP) [137]. ePARP is a combination of endoscopic identification of the lower rectal pouch, endoscopic assisted transperineal puncture and dilation of the new rectal tract and then a modified pull through of the rectal mucosa with rectoperineal anastomosis forming the neoanus. Although not commonly performed yet, this endoscopic assisted surgical approach may offer less trauma to the anal sphincter complex than the current surgical approach.

#### 3.8.6. Robots, SILS and NOTES, the Ideal Combination?

With the technical advancement of robotics, namely smaller diameter and wristed instruments, the combination of single incision surgery or NOTES with robotics appears to open a new era of robotically assisted single port or NOTES procedures, which were not able with the until then available laparoscopic instruments [138].

The SPORT platform (Titan Medical) is a single port robotic system providing bimanual instruments, a camera and light access all through one incision [139]. This system is still under development but has been evaluated in a porcine model.

A similar approach is followed by Intuitive with the SP platform, which offers bimanual wristed instruments, lighting and camera through on incision with a diameter of 2.5 cm [140].

Both systems require incisions of at least 2.5 cm length and therefore appear not suitable in small children and infants, although any preclinical testing is pending.

#### 3.8.7. Artificial Intelligence and Augmented Reality in Pediatric Surgery

Augmented reality (AR) and artificial intelligence (AI) have already grown into our daily lives, as we are used to playing AR games on our mobile phones and AI assists in facial recognition for serious or fun applications [141]. It is therefore just a matter of time, when AR and AI will be implemented into minimally invasive surgical procedures [142]. Three dimensional computer assisted laparoscopy or robotic surgery is an excellent platform for combining data from medical imaging, such as preoperative CT scans or intraoperative ultrasound, with the actual surgical field displayed to the surgeon in terms of augmented reality thereby displaying subsurface structures not visible to conventional laparoscopy [143,144]. Recordings from thousands of procedures, for example cholecystectomies, can be analyzed by AI and will give a real time feedback to the operating surgeon, of where to find the delicate structures not to be damaged. AR and AI have been evaluated in many fields of surgery, any application in pediatric surgery is not yet established [145].

#### 3.8.8. The Artificial Womb

At first sight, the development of an artificial womb does not imply minimally invasive pediatric surgery [146]. However, many congenital malformations, such as myelomeningocele, congenital diaphragmatic hernia, gastroschisis, sacrococcygeal teratoma or congenital pulmonary malformations often affect the fetus prenatally and severely, leading to either hydrops fetalis with imminent fetal demise or irreparable damage of organs at birth. Any fetal surgical intervention leads to a massive trauma, not only the fetus, but also the uterine environment and mother, often resulting in preterm labor with the effect of adding neonatal prematurity to the malformation. With the advent of an artificial uterus, a whole new perspective opens up of early fetal intervention for specific malformations: The fetus can be transferred into an artificial uterus, removing the mother and maternal uterus from damage and preterm labor and therefore the fetus from probable prematurity. The fetus can much easier be operated on in specifically designed artificial environments and then let to be grown inside this artificial uterus until term. Although not applicable in humans yet, animal studies show very promising results. We deem—“extra-uterine intra artificial-uterine fetal surgery”—to be the natural evolution of this approach and the next logical step in the minimally invasive surgical management of congenital malformations.

### 3.9. In the End: Hard Data on Minimally Invasive Surgery

Patient, parental and caregiver bias is the most relevant factor in surgical research as most outcome parameters depend on patient self-awareness and caregivers’ perceptions as well as their expectations on the surgical technique used. As the term “minimally invasive surgery” implies, they expect MIS to be truly minimally invasive and therefore mobilize themselves and start oral diets earlier or perceive pain less painful, because they have been told to be operated on by minimally invasive procedures. Therefore, any study aiming to generate hard data on minimally invasive surgery has to be randomized, prospective and, most important, double blinded, at least during the short term of the hospital stay.

Looking at those studies, some can be found examining the effect of minimally invasive surgery with remarkable results.

For appendectomy, prospective randomized double blind trials in adults found no significant advantage of the laparoscopic compared to the open procedure for the postoperative course, complications, pain or lost workdays. Operating room costs and time were increased and the hospital stay was not shortened. Only quality of life scores at 2 weeks were in favor of the laparoscopic procedure [147,148,149,150]. An umbrella review of meta analyses reported a lower rate of surgical site infections but higher rate of intra-abdominal abscess formation in laparoscopic compared to open appendectomy [151]. Similar results were found in a meta-analysis of randomized controlled trials in children [152,153,154].

For Weber-Ramstedt pyloromyotomy in hypertrophic pyloric stenosis, one prospective randomized multicenter double blinded trial can be found. It reported a significantly faster time to full enteral feedings for the laparoscopic procedure (23.9 versus 18.5 h) and earlier hospital discharge (43.8 versus 33.6 h) although the time to first enteral feedings was not different between the two groups [155]. The overall complication rate was not different but the rate of intraoperative mucosal perforation and incomplete pyloromyotomy, the most relevant complications of the procedure, appeared higher in the laparoscopic group. Unfortunately, this report does not describe the method of blinding the patient’s mother, therefore leaving room for interpretation and thus limiting the study. Similar results to this interpretation, no decrease of the incidence of postoperative vomiting, a similar complication rate and risk of inadequate pyloromyotomy were reported by a prospective randomized but nonblinded trials [156,157].

Data of meta-analyses were presented for several other pediatric surgical conditions in a recent report [2]. Although relying on nonblinded trials, the authors concluded that the advantages of minimally invasive surgery (mainly time to enteral feedings and hospital stay) seem to outnumber the disadvantages, such as procedure specific complications.

Most analyses focus on soft outcome parameters of minimally invasive surgery, such as time to enteral feedings, time to full mobilization, duration of hospital stay, cosmetics of the wounds and quality of life, which often result in favor of minimally invasive surgery, partially because of the abovementioned bias [2]. Solid outcome parameters, such as intraoperative and postoperative complication rates as well as long term sequelae of the procedures are often just as reported as byproducts. This changes, when it comes to comparing minimally invasive with open surgery in surgical oncology. Due to the lack of hard data and mostly relying on non-randomized prospective or retrospective analyses, the feasibility and oncological safety of minimally invasive procedures are not proven in many fields of surgery, including pediatric surgery. In 2018, the reports of Ramirez and Melamed, published in the New England of Surgery, changed the way many gynecologists approach early-stage cervical cancer, as they were able to demonstrate that minimally invasive radical hysterectomy was associated with lower rates of disease-free survival and overall survival than open abdominal radical hysterectomy [158,159].

Hard data, such as prospective randomized and double-blind studies, are lacking in pediatric surgical oncology. It is therefore of utmost importance that radical resection and oncological safety should never be jeopardized against soft outcome parameters of minimally invasive surgery.

## 4. Conclusions

Minimally invasive pediatric surgery is just evolving. Remotely controlled, through natural orifices deployed mini robots or single incision robotic assisted surgery with microinstruments of less than 3 mm diameter, augmented reality in combination with three dimensional stereoscopic view or ex utero in artificial utero fetal surgery all promise to increase the surgical benefit and reduce the surgical trauma of our patients.

However, when it comes down to hard data, as reported in randomized prospective double blinded trials, the so often proclaimed advantages of minimally invasive surgery in children become less evident. Thus, patient and parental counseling must always include all surgical and non-surgical options and—when medically justifiable—include their personal opinion into the decision process. Furthermore, all pediatric surgeons should strive to generate much more hard data on minimally invasive surgery by conducting or participating in randomized controlled double blinded trials.

## Figures and Tables

**Figure 1 jcm-09-03999-f001:**
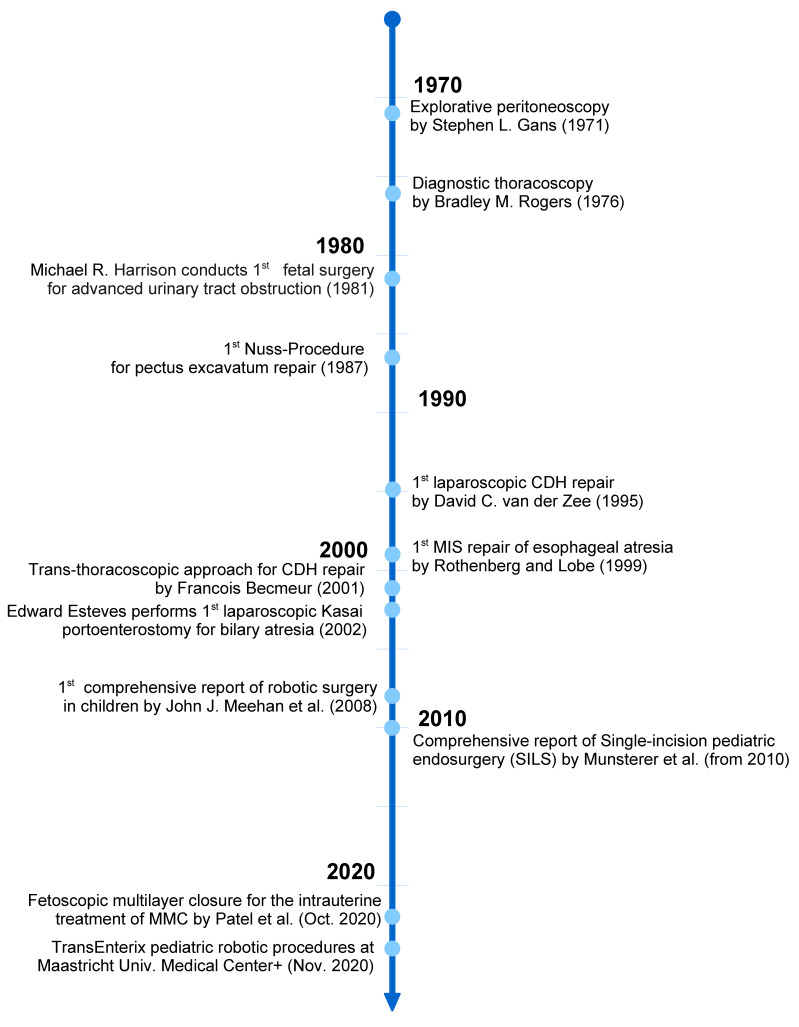
Timeline of the milestones of minimally invasive surgery.

**Figure 2 jcm-09-03999-f002:**
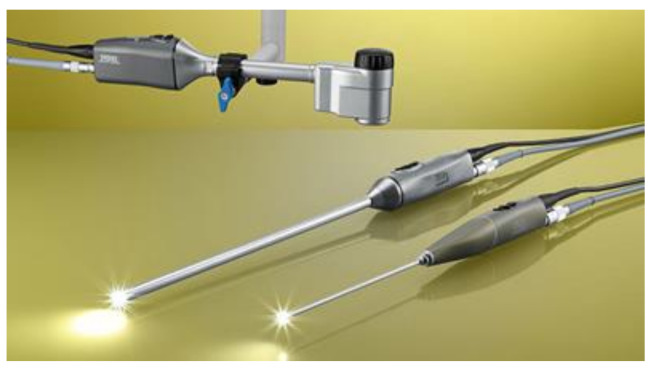
IMAGE1 STM D3-LINK Module with TIPCAM^®^1 STM 3D, 10 and 4 mm diameter, available with 0° and 30° optic. The 4 mm 3D optic combines three dimensional vision with a small diameter access especially in infants. Taken from: https://www.karlstorz.com [23].

**Figure 3 jcm-09-03999-f003:**
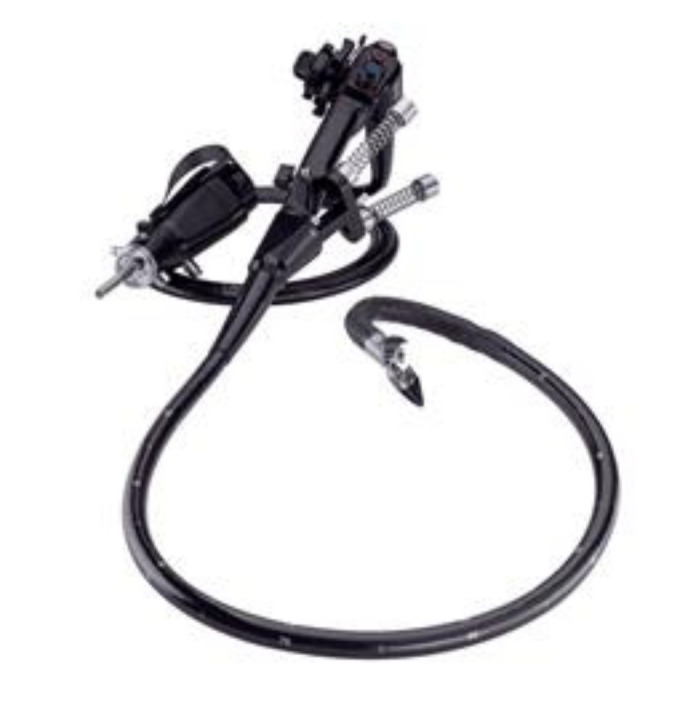
The ANUBIS project from Karl Storz for NOTES endoluminal and transluminal procedures [31].

**Figure 4 jcm-09-03999-f004:**
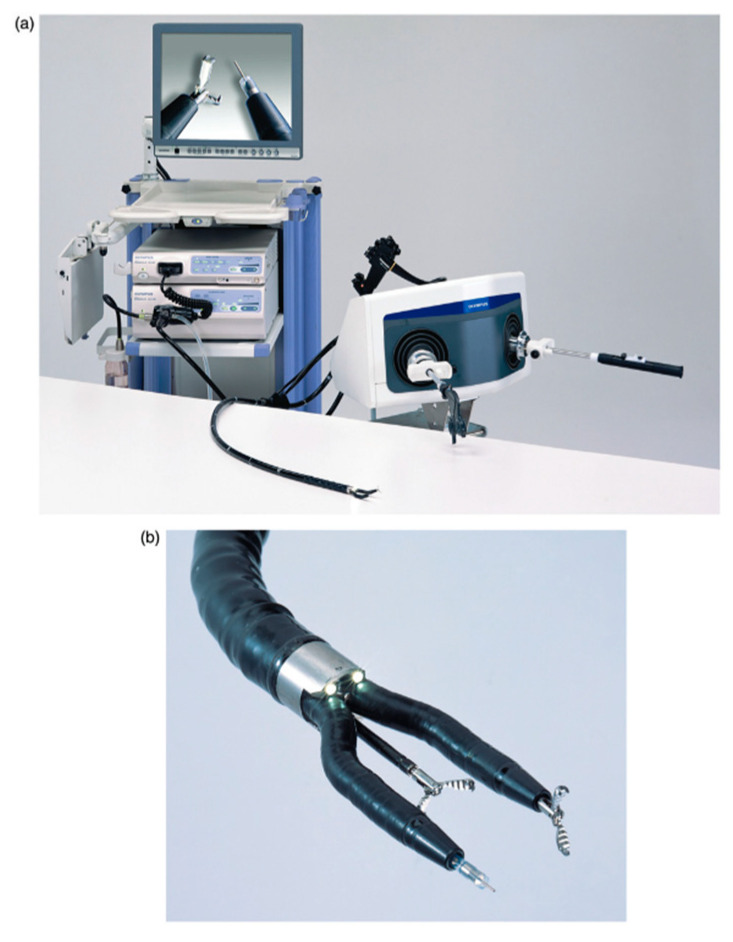
The ENDOSAMURAI by Olympus is a platform with two arms fitted to the tip of an endoscope that includes forceps channels in addition to the endoscope itself and its two arms. (**a**) The platform and (**b**) closeup of the tip of the instrument. Image taken from Kume, 2016 [28].

**Figure 5 jcm-09-03999-f005:**
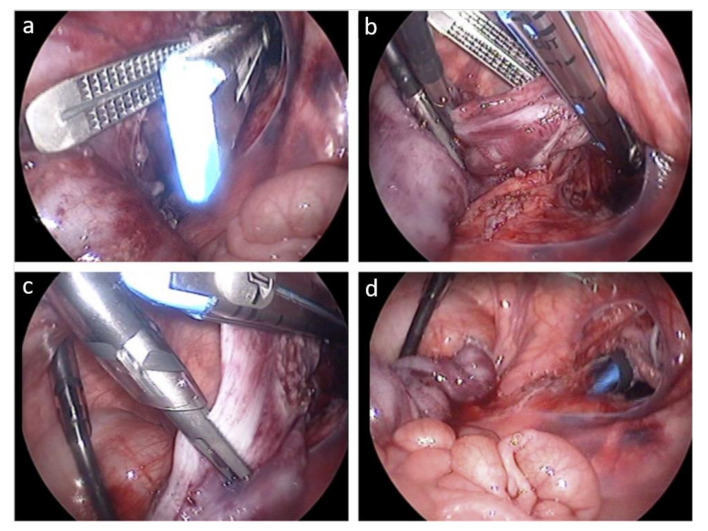
Laparoscopic Duhamel procedure assisted by transrectal NOTES. Three 5 mm transabdominal trocars were combined with a 12 mm transrectal trocar. (**a**) Introduction of the endoscopic stapler, (**b**) adjusting the stapler, (**c**) stapling of the colon, (**d**) situs after transection of the colon. Figure taken from Lamas-Pinheiro 2012 [34].

**Figure 6 jcm-09-03999-f006:**
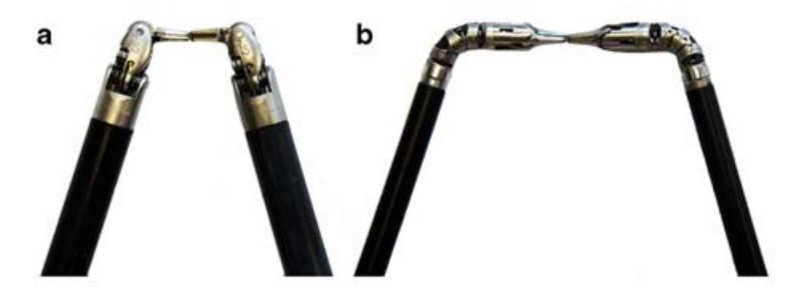
Comparison of the DaVinci 8 mm (**a**) and 5 mm diameter instruments (**b**). Due to their tentacle-like continuum tool shafts, the smaller diameter instruments need more operative space than the 8 mm instruments and are therefore not suitable in smaller cavities, such as infants. Figure taken from Marcus 2015 [62].

**Figure 7 jcm-09-03999-f007:**
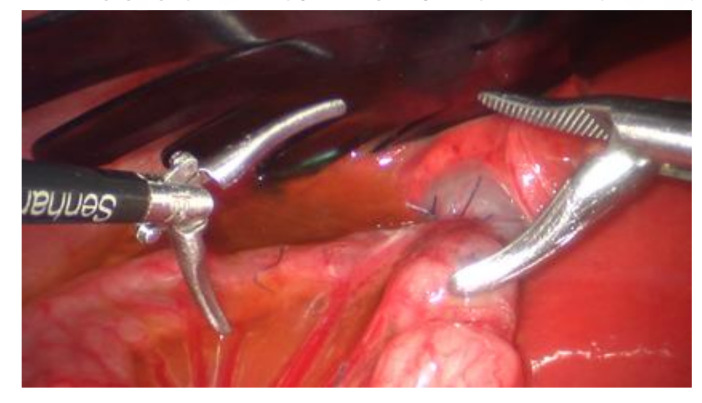
Robotic assisted (Senhance, Transenterix) cholecystoenterostomy in a 5 kg piglet with 3 mm and 5 mm instruments. The size of the instruments compared to the gallbladder and intestine demonstrates the small size of the cavity in which it is being operated.

**Figure 8 jcm-09-03999-f008:**
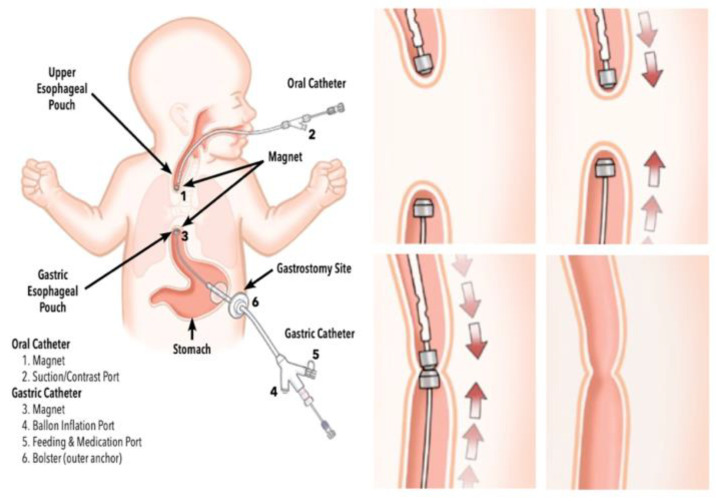
FDA-approved device for magnetic compression anastomosis in infants with long gap esophageal atresia. Magnamosis established by the Flourish™ device. It consists of two catheters, each holding a magnet at its tip. The magnets placed at each end of the esophagus attract each other, causing the ends of the esophagus to stretch toward each other and eventually creating an anastomosis with the open passage of the esophagus. Figure taken from Morrow 2017 and www.cookmedical.com [93,96].

**Figure 9 jcm-09-03999-f009:**
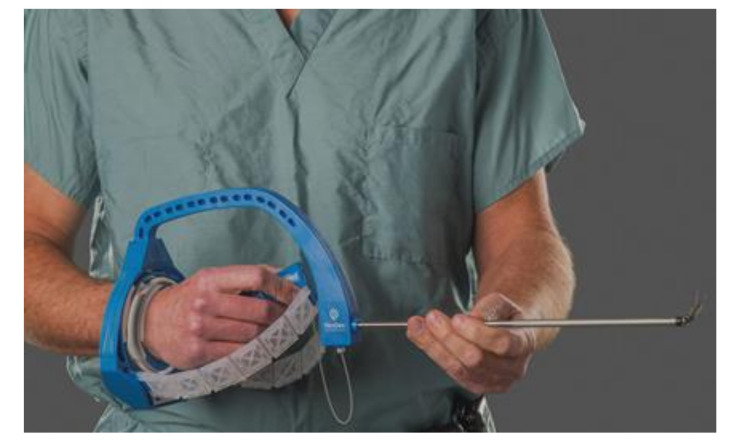
The FlexDex laparoscopic needle driver offers wristed instrumentation similar to DaVinci robotic instruments without the costs of the robot, but still more expensive than conventional laparoscopic instruments. Image taken from https://flexdex.com [99].

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
