# Peer review of "Advances and Trends in Pediatric Minimally Invasive Surgery"

_jcm, 2020, doi:10.3390/jcm9123999_

Round 1

Reviewer 1 Report

This is an excellent review on laparoscopic surgery in children. The report adds nothing new to all that can be found in the literature. On the other hand, this is an excellent review telling the story of and the state of the minimally invacive surgery to day and ideas on future development. As such this is an enjoyable reading to pediaric surgerons.

Author Response

Dear Reviewer 1,

Dear Madam or Sir,

            thank you very much for reviewing our manuscript. Please find a point-by-point response to your comments:

This is an excellent review on laparoscopic surgery in children.

Thank you very much.

The report adds nothing new to all that can be found in the literature.

Thank you for this comment. We aimed at summarizing in a comprehensive way the current state and trends in pediatric surgery as available in the recent literature. Nevertheless, concerning the outlook or perspective of minimally invasive surgery, we think the concept of ex utero intra artificial uterine fetal surgery has not been reported this extensively.

On the other hand, this is an excellent review telling the story of and the state of the minimally invacive surgery to day and ideas on future development.

Thank you very much!

As such this is an enjoyable reading to pediaric surgerons.

Thank you again very much, we appreciate your comments!

We would be very happy if you accept our paper for publication, please feel free to send me an email concerning any further questions.

Sincerely,

Robert Bergholz, M.D.

Reviewer 2 Report

A very well done and thorough review.  I question whether the robotic laboratory advancements in the timeline really fit.  No one has used the anastomotic robot in practice and I dob't think it as worthy to be on timeline.

Author Response

Dear Reviewer 2,

Dear Madam or Sir,

            thank you very much for reviewing our manuscript. Please find a point-by-point response to your comments:

A very well done and thorough review.

Thank you very much!

I question whether the robotic laboratory advancements in the timeline really fit. No one has used the anastomotic robot in practice and I dob't think it as worthy to be on timeline.

We totally agree with you that those laboratory robots have not been used in practice. We included them to demonstrate the progress pediatric surgery is also making in robotic surgery. Nevertheless, this timeline should be seen as a timeline of clinical achievements and we therefore deleted those two laboratory advancements according to your suggestions.

Thank you again very much, we appreciate your comments!

We would be very happy if you accept our paper for publication, please feel free to send me an email concerning any further questions.

Sincerely,

Robert Bergholz, M.D.

Reviewer 3 Report

It is a good job overall. However, much space is devoted to the discussion of new robotic procedures instead of those already consolidated. in fact, if you aim at minimal invasiveness in the child, more importance should be given to less invasive and already codified techniques such as single incision

In the chapter "Single incision laparoscopic and thoracoscopic surgery", line 172, delete the last sentence "This shows that while single incision laparoscopic appendectomy is well discussed and reported as a safe and standardized procedure, other single incision procedures are still to be established by clinical trial" and put: "the single incision procedure is a versatile technique in children, providing a safe, effective, and the least invasive treatment for several different diseases" reference: Versatility of one-trocar surgery in children J Laparoendosc Adv Surg Tech A. 2011 Jul-Aug;21(6):549-54.

Author Response

Dear Reviewer 3,

Dear Madam or Sir,

            thank you very much for reviewing our manuscript. Please find a point-by-point response to your comments:

It is a good job overall.

Thank you very much.

However, much space is devoted to the discussion of new robotic procedures instead of those already consolidated.

Thank our very much for this important comment. Robotic pediatric surgery is already established with Intuitive’s DaVinci system, most common in pediatric urology but also general and thoracic surgery. With the series reported by Meehan1 and others, we consider DaVinci pediatric surgery as an established technique just as you.2 The only drawback of the DaVinci is its limitation in small spaces, as demonstrated by several colleagues of us.3–5 This is where we think the new and emerging robotic platforms, such as the Senhance, the upcoming robot by CMR, the Dexter from distalmotion, the robot by avatera or systems from Kawasaki and Verb Surgical might be able to fill the gap and enable us pediatric surgeons to offer robotic assisted surgery even to newborns and infants. These systems are what we consider to be currents trends in pediatric surgery. This is the reason we separated the issue robotic into “robotic pediatric surgery” with description of the currently available systems and “new robots for children” to describe the abovementioned potential trends.

in fact, if you aim at minimal invasiveness in the child, more importance should be given to less invasive and already codified techniques such as single incision

Thank you very much for this comment. You are absolutely right and we added and changed text in this chapter, please see also the answer to your last comment:

In the chapter "Single incision laparoscopic and thoracoscopic surgery", line 172, delete the last sentence "This shows that while single incision laparoscopic appendectomy is well discussed and reported as a safe and standardized procedure, other single incision procedures are still to be established by clinical trial" and put: "the single incision procedure is a versatile technique in children, providing a safe, effective, and the least invasive treatment for several different diseases" reference: Versatility of one-trocar surgery in children J Laparoendosc Adv Surg Tech A. 2011 Jul-Aug;21(6):549-54.

We deleted the last sentence and put: “Based on those available data, single incision laparoscopic or laparoscopic assisted surgery has for many pediatric surgeons emerged as the first choice of the mode of minimally invasive access in many different procedures. Conversion to classical three port laparoscopic or thoracic surgery can be easily performed when procedures were started through a single incision. Therefore, the single incision procedure is a versatile technique in children, providing a safe, effective, and the least invasive treatment for different diseases.[51]”

Thank you again very much, we appreciate your comments!

We would be very happy if you accept our paper for publication, please feel free to send me an email concerning any further questions.

Sincerely,

Robert Bergholz, M.D.

  1. Meehan JJ, Sandler A. Pediatric robotic surgery: A single-institutional review of the first 100 consecutive cases. Surg Endosc. 2008;22:177–182.
  2. George EI, Brand TC, LaPorta A, et al. Origins of Robotic Surgery: From Skepticism to Standard of Care. JSLS.;22 . Epub ahead of print December 2018. DOI: 10.4293/JSLS.2018.00039.
  3. Thakre AA, Bailly Y, Sun LW, et al. Is smaller workspace a limitation for robot performance in laparoscopy? J Urol. 2008;179:1138–1142; discussion 1142-1143.
  4. Ballouhey Q, Clermidi P, Cros J, et al. Comparison of 8 and 5 mm robotic instruments in small cavities : 5 or 8 mm robotic instruments for small cavities? Surg Endosc. 2018;32:1027–1034.
  5. Finkelstein JB, Levy AC, Silva MV, et al. How to decide which infant can have robotic surgery? Just do the math. J Pediatr Urol. 2015;11:170.e1–4.
